# Antioxidant Responses of Phenolic Compounds and Immobilization of Copper in *Imperata cylindrica*, a Plant with Potential Use for Bioremediation of Cu Contaminated Environments

**DOI:** 10.3390/plants9101397

**Published:** 2020-10-20

**Authors:** Catalina Vidal, Antonieta Ruiz, Javier Ortiz, Giovanni Larama, Rodrigo Perez, Christian Santander, Paulo Ademar Avelar Ferreira, Pablo Cornejo

**Affiliations:** 1Centro de Investigación en Micorrizas y Sustentabilidad Agroambiental, CIMYSA, Universidad de La Frontera, Avda. Francisco Salazar, 01145 Temuco, Chile; catalinavidalp@gmail.com (C.V.); maria.ruiz@ufrontera.cl (A.R.); rodrigo.esteban.perez@gmail.com (R.P.); c.santander01@ufromail.cl (C.S.); 2Programa de Doctorado en Ciencias de Recursos Naturales, Universidad de La Frontera, Avda. Francisco Salazar, 01145 Temuco, Chile; 3Laboratorio de Biorremediación, Facultad de Ciencias Agropecuarias y Forestales, Universidad de La Frontera, Avda. Francisco Salazar, 01145 Temuco, Chile; j.ortiz01@ufromail.cl; 4Centro de Modelación y Computación Científica, Universidad de La Frontera, Avda. Francisco Salazar, 01145 Temuco, Chile; giovanni.larama@ufrontera.cl; 5Departamento de Solos, Universidade Federal de Santa Maria, Santa Maria, Rio Grande do Sul 97105-900, Brazil; ferreira.aap@gmail.com

**Keywords:** bioaccumulation, copper accumulation, copper mining, metallophyte, phenolic compounds

## Abstract

This work examined the capability of *Imperata cylindrica* to respond, tolerate and accumulate Cu when growing at high Cu concentration (300 mg kg^−1^ of substrate) at different times of exposure (2, 14 and 21 days). The Cu accumulation in plants was examined by atomic absorption spectroscopy (AAS) and Cu localized by Scanning Electron Microscopy-Energy Dispersive X-Ray spectroscopy. Additionally, the phenolic compound identifications and concentrations were determined using liquid chromatography coupled to mass spectrometry. Our results showed that root biomass decreased significantly at high Cu levels, with a greater decrease at 21 days (39.8% less biomass in comparison to control). The root showed 328 mg Cu kg^−1^ dry weight at 21 days of exposure to Cu, being the tissue that accumulates most of the Cu. Lipid peroxidation was a clear indicator of Cu stress, principally in shoots. The exposure to Cu significantly increased the synthesis of phenolic compounds in shoots of plants exposed 21 days to Cu, where 5-caffeoylquinic acid reached the highest concentrations. Our results support that *I. cylindrica* is a Cu accumulator plant in root organs with a medium level of accumulation (between 200–600 mg Cu kg^−1^ biomass), which can tolerate the exposure to high Cu levels by means of increasing the synthesis of phenolic compound in shoots, suggesting a potential use as phytoremediation tool in Cu polluted environments.

## 1. Introduction

It is well known that mining is one of the most invasive activities causing environmental pollution since it produces a large accumulation of potentially toxic elements (PTEs) in soils [1,2]. As an example, in Chile copper (Cu) mining has generated over-accumulation of this element in areas surrounding Cu smelters, being one of the most available PTEs in these soils [3]. Although Cu is an essential micronutrient for plants, when it exceeds the limits required by the plant it can be toxic for cellular functioning, principally by Cu-induced increases in the tissue oxidation causing a multitude of nocuous effects in the main macromolecules (membranes, nucleic acids and proteins) and even can induce cellular death [4,5]. The average concentration of Cu necessary for the proper functioning of plants ranges from 5 to 30 mg kg^−1^ dry weight; however, these values depend on some factors such as the characteristics of the soil where the plant is growing [6,7]. When Cu concentration in the tissue is over these limits, the effects of Cu toxicity become visible and several cellular processes are affected in the plant. At the landscape level, the Cu toxicity can generate losses of plant diversity and functionality, leading to changes in soil characteristics and avoiding the plant recruitment and establishment [8,9].

It is known that prolonged times of exposure to PTE triggers a variety of physico-chemical alterations in plant metabolism [10], causing indirectly a selection of resistant/tolerant populations, which develop diverse strategies of exclusion, accumulation, and stabilization of the PTE in their tissues, representing an adaptive advantage that makes them candidates for further uses in the development of bioremediation techniques for PTE-contaminated environments.

Among the main intracellular mechanisms involved in reducing the amount of metal, we highlight the chelation of PTE through amino acids, organic acids, glutathione (GSH), or PTE-binding ligands, compartmentation within vacuoles, and upregulation of the antioxidant response. In the latter, phenolic compounds are characterized by having various functions in plants, highlighting its antioxidant capacity observed under different stress conditions [11]. Exposure to PTEs generates reactive oxygen species (ROS) and increases the production of phenolic compounds in plants [12]. Phenols act as chelating metals through hydroxyl and carboxyl groups and also inhibit lipid peroxidation by trapping alkoxyl radicals [13]. Therefore, phenolic compounds exert their effects by scavenging ROS or avoiding their generation [14,15]. The scavenging ROS activity of phenolic compounds are determined by their chemical structure, type and number of functional groups [16].

*Imperata cylindrica* is a grass with a wide geographical distribution including places with high concentrations of metals in soil [17]. This plant species also inhabits the central Valley of the Valparaiso Region, Chile, place recognized by the great environmental pollution originating mainly by the Cu smelting. Studies in *I. cylindrica* from Río Tinto (Huelva, Spain), classified it as an iron (Fe) hyperaccumulator, finding concentrations of up to 2.3% in its tissues, mainly in the root [18,19]. Nevertheless, at the date, there are very few antecedents about its capacity such as Cu accumulator, despite its wide distribution in Cu-contaminated soils. Regarding its antioxidant activity mediated by phenolic compounds, only two compounds have been identified from the shoots; trans-p-coumaric acid and tricin [20]. However, there are no studies evidencing its response, including phenolic compounds, to Cu stress. As this species is a common inhabitant in strongly polluted environmental [8,21], new antecedents could be useful to characterize other plants species to be used in bioremediation. Therefore, the objective of this work was to evaluate the responses of *I. cylindrica* under Cu stress and establish whether phenolic compounds play an important role in the antioxidant response against Cu, together with the capability to accumulate Cu in its tissues.

## 2. Results

### 2.1. Biomass Production

The interaction between the factors time and Cu supply showed no significant difference for the shoot or root dry matter production (Figure 1A), being only responsive to the applied Cu doses. The addition of 300 mg kg^−1^ of Cu reduces the production of root dry matter in 30% compared to treatment without application of Cu at 21 days (Figure 1A). However, the Cu application to the substrate of *I. cylindrica* did not generate a significant effect on the production of shoot dry matter (Figure 1A).

### 2.2. Lipid Peroxidation

The exposure to Cu has a higher effect in the shoot lipid peroxidation than in roots (Figure 1B), producing a strong increase in the malondialdehyde (MDA) content for the three Cu exposure times, compared to the controls, ranging from 43% at 2 days and reaching a 108% of increase at 21 days (Figure 1B). In addition, there was a significant increase in the amount of MDA at 14 and 21 days with respect to two days of metal exposure. Meanwhile, the roots of *I. cylindrica* only showed an increase in MDA levels at 14 and 21 days of Cu exposure, with 80.3 and 84 nmol MDA g^−1^ fresh weight (FW), respectively (Figure 1B).

### 2.3. Cu Concentration in Plant and Cu Available in Substrate

There was no significant interaction between the factors for the Cu content in the shoots and roots. The Cu content in the shoots was 2.8 times higher in plants grown in the treatment with 300 mg of Cu kg^−1^ (Figure 2A). Meanwhile, in the roots of plants cultivated in the treatment with addition of Cu the contents of Cu were 9 times higher than in the control treatment, at 21 days after Cu exposure (Figure 2A). Additionally, the highest levels of Cu in the roots were observed at 21 days after Cu supply in the treatment, with 328 mg Cu kg^−1^ dry weight (Figure 2A). In the treatment without Cu application and in the treatment with 300 mg Cu regardless of the exposure time, 80% and 94% on average Cu was retained in the root system of the plants. Meanwhile, the electronic microscopy images coupled to energy-dispersive x-ray (EDX) only were able to evidence the Cu presence in the rhizome of plants exposed to Cu (Figure 3H), apparently in the surface of cell walls into the vascular system. The presence of Cu was unappreciable by the used methods both in non-Cu supplied treatments (Figure 3G) as well as in the shoot organs (Figure 3C,D).

The Cu available in the substrate showed low concentrations in the controls, between 1.43–1.54 mg Cu kg^−1^ substrate, while in the treatments with Cu addition were observed values between 95.4–121.2 mg Cu kg^−1^ substrate and a significant decrease in the metal content, especially at latter sampling stages (Figure 2B).

### 2.4. Quantification and Identification of Phenolic Compounds in Shoot

A total of ten phenolic compounds were registered and quantified (Appendix A), of which six could be tentatively identified by mass spectrometry, corresponding to two hydroxycinnamic acid derivatives: 5-caffeoylquinic acid (peak **1**) and caffeoylquinic acid isomer (peak **2**); three anthocyanin: cyanidin-3-hexoside (peak **3**), cyanidin-3-malonylglucoside (peak **5**) and cyanidin-derivative (peak **7**); and one flavon: orientin (peak **4**) (Table 1). The other four compounds (peaks **6**, **8**, **9A**, **9B** and **10**) according to their spectroscopic information as Diode-Array Detection (DAD) spectra and fragmentation pattern seems to be phenolic compounds but could not be identified. In addition, in the majority of compounds the concentrations were higher in the Cu treatment, respect to control, except for three unidentified compounds and caffeoylquinic acid isomer that showed no significant differences (Table 2). Moreover, 5-caffeoylquinic acid was the compound identified in higher concentrations both in the treatment as in the control with values of 1715 µg g^−1^ FW and 1387 µg g^−1^ FW, respectively (Table 2).

### 2.5. Total Phenols in Shoot and Antioxidant Activity

The content of total phenols measured by the Folin Ciocalteau method evidenced a significantly higher accumulation of these compounds in the shoot of *I. cylindrica* when growing under Cu stress (2.87 mg g^−1^ FW in the control and 3.4 mg g^−1^ of FW in the treatment) (Figure 4A). Meanwhile, the sum of the phenolic compounds identified by chromatography showed the same significant difference as the quantification by Folin Ciocalteau method (Table 2), where the differences between the concentrations determined by HPLC and spectrophotometric method could be explained for the presence of polymeric compounds. Antioxidant activity by DPPH in shoots of *I. cylindrica* after 21 days of Cu exposure did not show significant changes (Figure 4B). Otherwise, the antioxidant capacity by CUPRAC showed greater activity in the treatment with Cu applied, reaching 0.811 μmol g^−1^ TE compared to 0.663 µmol g^−1^ TE in the control (Figure 4B).

### 2.6. Multivariate Analysis

The principal components (PC) analysis evidenced the formation of two highly independent groups in accordance with the distribution of experimental variables in the two PCs. Detailing, PC1 explained 63.1% and PC2 12.2% of the total experimental variance (Figure 5). PC1 was positively influenced by the Cu content in plant and substrate, lipid peroxidation, antioxidant activities (DPPH and CUPRAC), the production of total phenols by both Folin and HPLC methods, and all the phenolic compounds quantified, with the exception of compound **9A** and **9B**. Otherwise, it was negatively related only to biomass production, and compound **9A** and **9B** production. PC2 was also positively related with Cu content in plant and substrate, antioxidant activities (DPPH and CUPRAC), production of total phenols by Folin and HPLC, production biomass, production of 5-caffeoylquinic acid, caffeoylquinic acid isomer, compound 6, compound **9A** and **9B** and compound **10**. Regarding the two groups of experimental units, the first corresponded with the samples treated with 300 mg Cu kg^−1^, which are grouped together with the Cu accumulation in shoot and substrate, the synthesis of phenolic compounds and the antioxidant activity of these compounds. Meanwhile, the second group included the control samples, which were grouped together with the biomass production.

## 3. Discussion

Plants can produce alterations in the root system in response to high levels of Cu in the soil solution. The root constitutes the first line of defense against Cu in the rhizosphere, and the initial characterization of Cu toxicity is the hindrance of root elongation and growth [22]. Higher Cu concentration can reduce the rate of cell division in the root apex, consequently reducing root length and increasing the number of lateral roots, impairing all root growth [23,24]. Here, we clearly appreciated that from the 2nd day of Cu supply there was a negative effect on the root growth, which was reflected in a reduced production of biomass. This effect was maintained in the following days of Cu exposure (14 and 21), further accentuating the difference respect to the control non-Cu supplied. Therefore, the root growth could be a better predictor of Cu damage than shoot biomass. The reduction of root growth can be changed by the root exudation of organic ligands (phenolic compounds, organic acids, phytosiderophores) that can complex with the free Cu species in solution reducing Cu availability to plants absorption [21,25,26], changing the plant defense mechanism.

Plant cells have evolved a myriad of adaptive mechanisms to manage excess of metal ions and utilize detoxification mechanisms to prevent their participations in unwanted toxic reactions. Initially, plants prevent or reduce the Cu uptake by restricting the metal ion flux from rhizosphere to the apoplast through binding them to the cell wall, chelating to cellular exudates or by inhibiting long distance transport [27,28]. However, when present at elevated concentrations, cells activate a complex network of storage and detoxification strategies, such as chelation of metal ions with phytochelatins and metallothioneins into the cytosol, trafficking, and sequestration into the vacuole by influx trough membrane transporters [29].

The results of Cu content in the plant showed that most of Cu accumulated in *I. cylindrica* is stored in root organs, and this process of uptake occurs within a few hours after the contact with the Cu ions. Besides, the Cu content in the root increases across time (Figure 2B), reaching values that allows classifying *I. cylindrica* as a potential phytostabilizer of Cu, with medium accumulation (200–600 mg Cu kg^−1^) in root organs [30]. This accumulation response has been widely observed in other plant species such as *Chloris gayana*, *Cucumis sativum* and some *Eucalyptus* species exposed to Cu, in which higher Cu accumulation in root than shoot is probably due to a fast absorption by the roots and its slow translocation to shoot [31]. Moreover, this behavior is similar to other species that cohabit with *I. cylindrica* in the Cu-ontaminated environment from Puchuncaví Valley, as *Oenothera picensis* [8,32,33,34]. Also, previous studies have shown how *I. cylindrica* inhabits soils with more than 300 mg kg^−1^ of Cu available [8] and how its accumulating capacity in the root increases according to the amount of Cu applied to the soil, at least between a range of 450–780 mg kg^−1^ of Cu in soil [35]. Although these reference values are a first approximation, the range of Cu levels in which *I. cylindrica* can be used in bioremediation seems to be much broader than the explored so far. Besides, data from Meier et al. [35], showed that its combined use with arbuscular mycorrhizal fungi can increase their bioaccumulation in root. These antecedents make evident the existence of mechanisms of Cu stabilization or accumulation in the *I. cylindrica* root organs, which deserve to be deeply analyzed to allow its continuous grow in an environment with noticeable Cu amounts, especially if bioremediation programs are planned. On the other hand, the Cu available in substrate support the observed rate of plant Cu accumulation, since over the days there is less Cu available in substrate, concomitantly with an increase of Cu in root tissues.

Toxic concentrations of Cu are well recognized for the increase of redox processes, which triggers the formation of ROS and could become attached to the sulfhydryl groups of cell membrane or induce lipid peroxidation resulting in the membrane damage and the production of free radicals in different plant organelles [36]. The measurement of MDA, a product of lipid peroxidation, is routinely used as an index of lipid peroxidation under stress conditions [37]. In this case, the Cu addition to growth substrate of *I. cylindrica* originated a clear increase of the lipid peroxidation in shoots and this effect was present in all the sampling times. Otherwise, in the root there was only an effect at 14 days after applying Cu, which would indicate that the shoot is more susceptible to the Cu presence than the root. Thereby, lipid peroxidation could be a good stress indicator for Cu in shoots, with a fast response.

Phenolic composition of root organs (mainly rhizomes) from *I. cylindrica* has been previously reported where mainly flavones were detected [38]; however, only p-coumaric acid and tricin have been previously reported in shoots [20]. It is noticeable that phenolic compounds as anthocyanins or hydroxycinnamic acids are main responsible of the antioxidant activity [39], also playing an important role in the protection of tissues to cope with the damage induced by ROS. As previously mentioned, here we identified six signals of phenolic compounds; however, according to the reported by Yang et al. [40], the other not-identified metabolites corresponded to flavonoid glycosydes (subtypes of flavonoid O-glycosides, detected in peaks **6**, **8**, **9A**, **9B** and **10**). Interestingly, the 5-caffeoylquinic acid presented a high production with respect to the other phenolic compounds analyzed. This chlorogenic acid is also one of the most abundant acids in the green fruit of coffee, considered as a good source of phenolic acids [41]. Moreover, the chlorogenic acids (5-caffeoylquinic acid and caffeoylquinic acid isomer) belongs to the group of non-colored compounds, which respond to measurements as CUPRAC, where there was a greater antioxidant activity of this type of compounds in plants under Cu stress. Among the compounds identified, cyanidins were also an important group. This group of anthocyanins have been described as responsible for the red coloring on leaves of diverse higher plants [42,43], probably playing the same role in the evident reddish coloration in shoots of *I. cylindrica*. At the same time, anthocyanins are pigments recognized for their protective role against UV radiation and excessive light, and it has been observed that their biosynthesis is stimulated by environmental stresses such as drought, attack by pathogens and insects [44]. In this study, Cu supply also generated an overproduction of cyanidines in shoots, suggesting that effectively *I. cylindrica* uses the biosynthesis of these compounds as a mechanism of protection against oxidative stress. Excessive concentrations of Cu in plant tissues influence the photosynthetic complex biosynthesis by reducing photosynthetic pigment content such as chlorophyll and carotenoids and change the chloroplast structure and thylakoid membrane composition [45,46]; therefore, the largest production of cyanidins can reduce this damaging effect to the photosynthetic complex caused by high concentrations of Cu.

Content of total phenols by both methods, Folin Cicolteau and HPLC-DAD, also showed a greater synthesis of such compounds in shoots when Cu was added to the substrate. Similar results were obtained by Sruthi and Puthur [47], who also reported an increase in the phenol content in *Brugueira cylindrica* exposed to increasing Cu concentrations. Additionally, the PC analysis clearly showed an association between treatment with Cu and the synthesis of phenolic compounds, demonstrating that in general these compounds would be a combat route against the stress produced by Cu in the metabolism of the plant. Therefore, plants with adaptations aimed to produce phenolic compounds in high amounts can be good candidates to be used in phytoremediation of Cu-contaminated environments, since only a few quantity of Cu hyper-accumulators plant species have been described at the time, which justify the preferential use of phytostabilization for Cu-contaminated environments.

## 4. Materials and Methods

### 4.1. Plant Collection and Experimental Design

Rhizomes of *I. cylindrica* were collected in the Puchuncaví Valley (Central Chile, 32°46′30′′ S, 71°28′17′′ W), Valparaíso Region, Chile, about 1.5 km from the Ventanas smelter. The rhizomes were cut and disinfected with 2% *w*/*v* chloramine-T solution for 5 min, then rinsed thoroughly with distilled water. Sterile-inert substrate was used for sprouting rhizomes, which was composed of sand and vermiculite (9:1; *v*/*v*). Plants were grown in a greenhouse for 6 months at 16/8 h light/dark photoperiod at 25 ± 3/15 ± 3 °C day/night temperatures, and watering with sterile distilled water. The substrate was washed and sterilized by autoclaving per three consecutive days and air-dried for 24 h. After sterilization, the sand/vermiculite mixture was supplemented with an equivalent to 200 mL of a solution 5.9 g L^−1^ of CuSO_4_·5H_2_O, which was allowed to equilibrate for 2 weeks at room temperature, as described by Aponte et al. [48]. This substrate represented the treatment with a nominal equivalent of 300 mg Cu kg^−1^. For the control treatment, 200 mL of distilled water was added to the substrate and let to equilibrate for 2 weeks. The assay compiled treatments with 300 mg Cu kg^−1^ substrate and controls without Cu addition, which were performed in pots with three plants per pot and four experimental units per treatment (*n* = 4). A basal fertilization of 18, 8 and 8 mg kg^−1^ of N, P, and K, respectively, was applied to all plots using a commercial fertilizer (Vitasac 18-8-8, Anasac Ambiental S.A., Región Metropolitana, Chile). After 6 months of growth the plants were incorporated to the described treatments. To measure the exposure time effect to Cu on the antioxidant activity in *I. cylindrica*, three sampling times were performed. The first at 2 days, the second at 14 days and the third at 21 days after the Cu application. The treatments were kept in a greenhouse under the conditions described above. After each sampling, plant roots were thoroughly rinsed in abundant deionized water. Afterwards, plants were separated into roots and shoots. Half of each tissue was stored at −80 °C for biochemical determinations and the other half dried at 60 °C in a forced air oven for 48 h and weighed.

### 4.2. Lipid Peroxidation

Lipid peroxidation was assayed according to the modified method of Du and Bramlage [49], based on the levels of MDA as determined by the reaction given by thiobarbituric acid. Briefly, to 0.15 g of fresh material of shoots and roots was added to 1 mL of 20% *v*/*v* trichloroacetic acid. Mixture was heated at 95 °C for 30 min, cooled quickly on ice and centrifuged at 10,000× *g* for 10 min. The reaction was measured in microplate spectrophotometer EPOCH (Bio Tek Instruments, Inc., Winooski, VT, USA) where the absorbance was read at 440, 532 and 600 nm.

### 4.3. Cu Concentration

Cu concentration in shoots and roots was measured using 1 g of dry plant material. The tissue was crushed and converted into ashes in a furnace at 550 °C for 12 h, then digested using a H_2_O/HCl/HNO_3_ mixture (8:1:1; *v*/*v*/*v*). The Cu content was determined by atomic absorption spectroscopy (AAS; Unicam SOLAAR, mod. 969, Cambridge, UK). Available Cu in the substrate was measured using 10 g of air-dried substrate with 20 mL of diethylenetriaminepentaacetic acid (DTPA) extractant solution (5 mM DTPA, 0.1 M triethanolamine and 10 mM CaCl_2_, pH 7.3). The mix was shaking for 2 h. After, the Cu in the extract was determined by AAS using external calibration.

### 4.4. Cu Localization in Plant Tissues

In order to localize the Cu-bound to the shoot and root tissues, plants with 21 days of growth were observed by Variable Pressure Scanning Electron Microscope (VP-SEM), with transmission module STEM SU-3500 (Hitachi, Tokyo, Japan). The presence of Cu was verified by Energy Dispersive X-Ray Spectrometer Detector (EDX), QUANTAX 100 (Bruker, Karlsruhe, Germany) with BSE detector in transversal sections of shoot and root (rhizome). The analyses were carried out in the Scientific and Technological Bioresource Nucleus (BIOREN), Universidad de La Frontera, Temuco, Chile.

### 4.5. Identification and Quantification of Phenolic Compounds

Half a gram of frozen shoots (−80 °C) was pulverized with liquid nitrogen and mixed with 1.5 mL of a solution of methanol/formic acid (97/3; *v*/*v*), subjected to an ultrasonic bar for 60 s, followed by orbital agitation for 15 min at room temperature. Then, a centrifugation step at 4000× *g* for 10 min was necessary to obtain the extract. All the extracts were dried in a rotary evaporator and re-suspended in 1 mL of the mobile phase (water:acetonitrile:formic acid; 92:3:5; *v*/*v*/*v*). Quantification of phenolic compounds was carried out in a Shimadzu HPLC system (Tokyo, Japan) equipped with a quaternary LC-20AT pump with a DGU-20A5R degassing unit, a CTO-20A oven, a SIL-20a autosampler and an UV-vis diode array spectrophotometer (SPD-M20A), according to the described by Santander et al. [50]. HPLC-DAD system coupled to a mass spectrometer (QTrap LC/MS/MS 3200 Applied Biosystem MDS Sciex system (Foster City, CA, USA)) was used to obtain the identity assignments [51].

### 4.6. Total Phenols and Antioxidant Activity Determinations

Total phenol concentration was determined by the Folin-Ciocalteu method as described by Singleton and Vitic [52], with minor modifications, in the same extract described above. Gallic acid was used as the standard. The reagents were added in the following order: 10 µL of gallic acid or sample, 0.5 mL of water, 50 µL of Folin–Ciocalteau reagent, 200 µL of 20% *w*/*v* sodium carbonate and completed to a volume of 1.5 mL with ultrapure water. The reaction was incubated to 20 °C by 30 min and then the absorbance was measured at 750 nm in microplate spectrophotometer EPOCH (BioTek Instruments, Inc., Winooski, VT, USA). Antioxidant activity was determined by Cupric Ion Reducing Antioxidant Capacity (CUPRAC) and 2,2-diphenyl-1-picrylhydrazyl (DPPH) methodologies, according to the described by Parada et al. [51]. The results were expressed as Trolox equivalents (TE). All antioxidant activities were performed in microplate spectrophotometer EPOCH (BioTek Instruments, Inc., Winooski, VT, USA).

### 4.7. Statistics

For the study, a completely randomized 3 × 2 factorial design with four replicates was established with three exposure times (2, 14 and 21 days), and without Cu or with 300 mg Cu kg^−1^. Data sets were analyzed by means of analysis of Variance (ANOVA) of one or two ways. Ln transformation was used when the data sets were not meeting the ANOVA assumptions (homoscedasticity and normality); nevertheless, all results are expressed in their original scale of measurement. Tukey’s multiple range test was used to compare the means of the treatments at 21 days (phenolic compounds). Subsequently, data sets were also subjected to principal component (PC) analyses. Variability in the means was expressed as the standard error and a *p* < 0.05 was considered statistically significant. The IBM SPSS statistic software v. 22.0 was used for all procedures.

## 5. Conclusions

The Cu presence across the time caused a decrease in the biomass production especially in root organs of *Imperata cylindrica*. Interestingly, this tissue exhibited a marked capability to accumulate Cu at medium-high rates, which can support its use in phytoremediation initiatives in Cu-contaminated environments. Cu also showed a strong effect on the lipid peroxidation of shoots, which was increased over time, this parameter being a good indicator of the Cu effect on shoots of *I. cylindrica*. On the other hand, the analysis of phenolic compounds in shoots revealed the presence of hydroxycinnamic acids, anthocyanins and flavons, compounds with great antioxidant capacity that demonstrate that *I. cylindrica* possess an interesting set of defense mechanism against Cu stress, probably allowing its use even at extreme Cu contamination levels.

## Figures and Tables

**Figure 1 plants-09-01397-f001:**
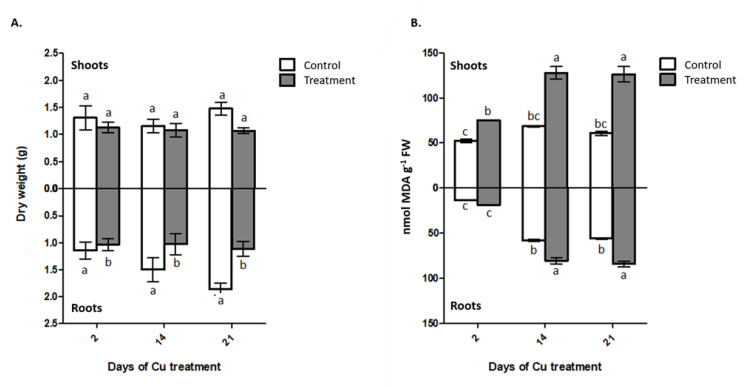
Dry matter and lipid peroxidation of *Imperata cylindrica.* (**A**) Dry matter production in shoots and roots; (**B**) Lipid peroxidation in shoots and roots. Where, control (without Cu applied) and treatment (300 mg Cu kg^−1^ substrate). Different lowercase letters indicate statistic difference according Tukey’s multiple range test (*p* < 0.05). Data are expressed as mean ± SE, *n* = 4.

**Figure 2 plants-09-01397-f002:**
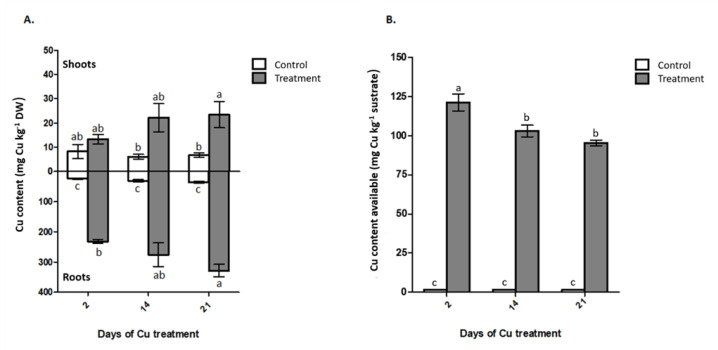
Copper concentrations. (**A**) Cu accumulation in shoots and roots of *Imperata cylindrical*; (**B**) Cu content available in substrate. Where, control (without Cu applied) and treatment (300 mg Cu kg^−1^ substrate. Different lowercase letters indicate statistic difference according Tukey’s multiple range test (*p* < 0.05). Data are expressed as mean ± SE, *n* = 4.

**Figure 3 plants-09-01397-f003:**
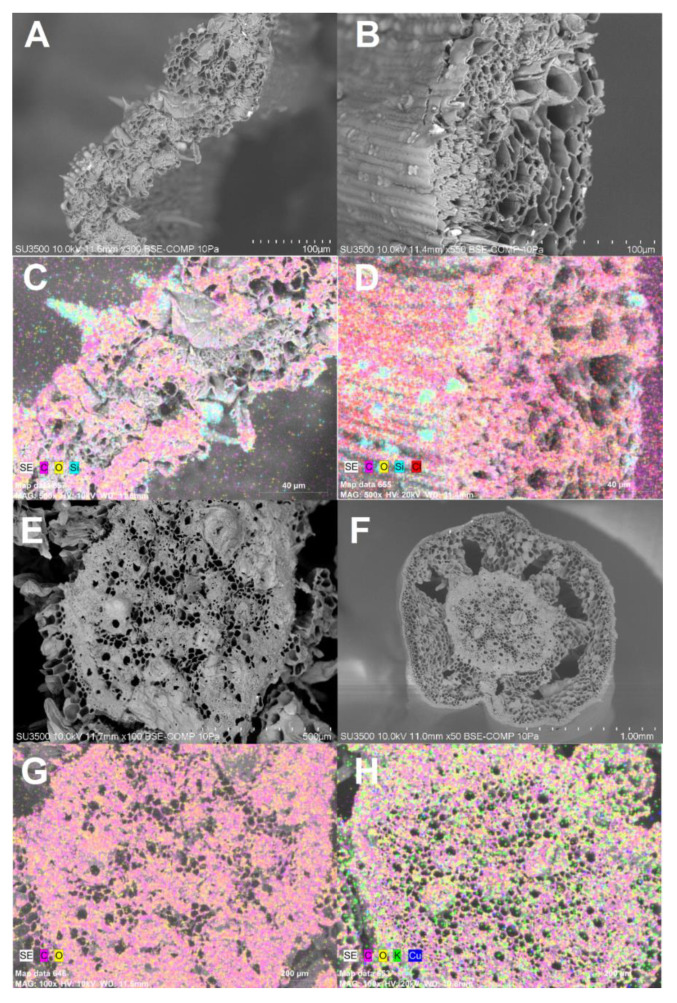
Scanning electron microscopy (SEM) and energy-dispersive x-ray (EDX) analysis of *Imperata cylindrica* after 21 days of growth. (**A**) Transverse cutting of shoot under control condition, and (**B**) Transverse cutting of shoot under stress condition (300 mg Cu kg^−1^ substrate). (**E**) Transverse cutting of root (rhizome) under control condition, and (**F**) Transverse cutting of root (rhizome) under stress condition (300 mg Cu kg^−1^ substrate). Elemental localization by EDX analysis: (**C**) Transverse cutting of shoot under control condition, and (**D**) Transverse cutting of shoot under stress condition (300 mg Cu kg^−1^ substrate). (**G**) Transverse cutting of root (rhizome) under control condition, and (**H**) Transverse cutting of root (rhizome) under stress condition (300 mg Cu kg^−1^ substrate).

**Figure 4 plants-09-01397-f004:**
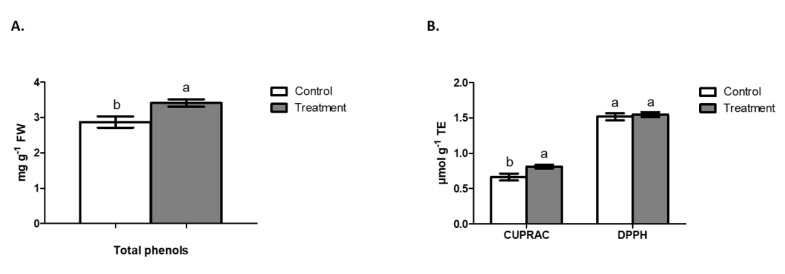
Total phenols and antioxidant activity in phenolic extract of *Imperata cylindrica* shoots. (**A**) Total phenols by Folin Ciocalteau, after 21 days of Cu exposure. (**B**) Antioxidant activity determination by DPPH and CUPRAC after 21 days of Cu exposure. Control condition (without Cu applied) and treatment (300 mg Cu kg^−1^ substrate). Different lowercase letters indicate statistic difference according Tukey’s multiple range test (*p* < 0.05). Data are expressed as mean ± SE, *n* = 4.

**Figure 5 plants-09-01397-f005:**
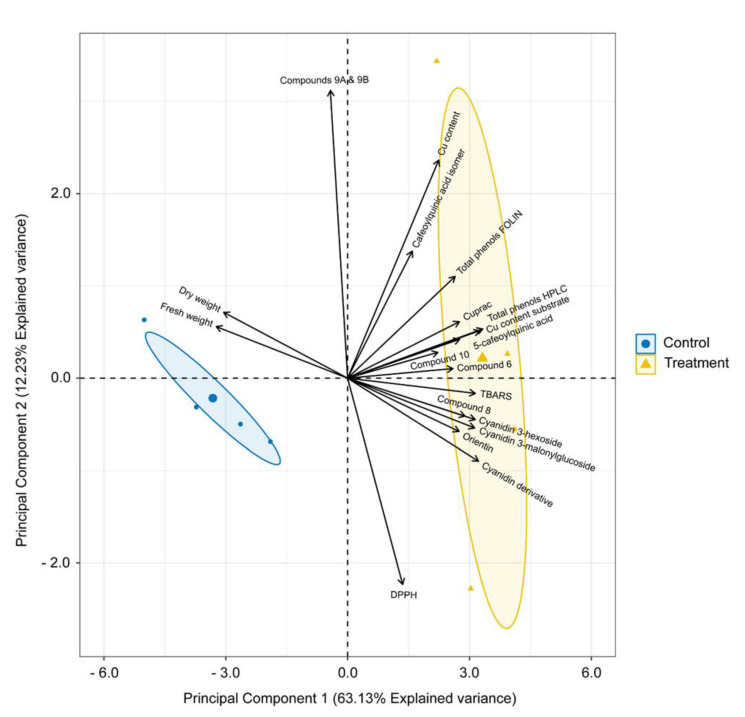
Principal component analysis (PCA) with scores for the respective association between Cu exposure during 21 days on *I. cylindrica* shoot and their variables analyzed. Control conditions (without Cu applied) and treatment (300 mg Cu kg^−1^ substrate). The mean value was used in each situation.

**Table 1 plants-09-01397-t001:** Identification of phenolic compounds from *Imperata cylindrica* shoots by HPLC-DAD-ESI-MS/MS after 21 days of growth.

Peak	t_R_ (min)	Group	Compound	ʎ max	[M + H]^+^	[M − H]^−^	Productions
**1**	5.5	HCADs	5-caffeoylquinic acid	324		353.5	191.2
**2**	6.6	HCADs	caffeoylquinic acid isomer	323		353.6	191.2
**3**	8.8	Anthocyanin	cyanidin-3-hexoside	515	449.8		287.5
**4**	12.5	Flavon	Orientin	348		447.6	357.3; 327.3; 430.3; 297.4; 287.3
**5**	12.6	Anthocyanin	cyanidin-3-malonylglucoside	516	535.7		287.2
**6**	13.4	* Flavonoid glycosides	*	348		549.7	400.3; 370.4; 460.4; 490.4; 430.3
**7**	14.0	Anthocyanin	cyanidin-derivative	517	621.8		287.5
**8**	15.7	* Flavonoid glycosides	*	345		417.6	358.3; 328.2; 400.3; 298.3
**9A**	18.1	* Flavonoid glycosides	*	348		575.7	326.2; 298.3; 412.3; 430.3
**9B**	18.3	* Flavonoid glycosides	*	349		575.7	412.3; 474.3; 298.3; 338.2; 309.2
**10**	19.3	* Flavonoid glycosides	*	346		431.6	358.3; 328.1; 298.2; 285.2

HCADs: Hydroxycinnamic acid derivatives. * Not identified.

**Table 2 plants-09-01397-t002:** Concentrations of phenolic compounds from *Imperata cylindrica* shoots by HPLC-DAD. Control condition (without Cu applied) and treatment (300 mg Cu kg^−1^ substrate).

N Compound	Phenolic Compound	µg g^−1^ FW	
		Control	Treatment
**1**	5-caffeoylquinic acid	1387.1 ± 107.1 ^b^	1715.0 ± 136.3 ^a^
**2**	caffeoylquinic acid isomer	217.6 ± 9.3 ^a^	247.1 ± 45.4 ^a^
**3**	cyanidin-3-hexoside	13.3 ± 2.3 ^b^	33.3 ± 7.3 ^a^
**4**	Orientin	242.6 ± 39.3 ^b^	395.5 ± 78.8 ^a^
**5**	cyanidin-3-malonylglucoside	30.9 ± 4.0 ^b^	72.3 ± 13.4 ^a^
**6**	*	113.7 ± 24.1 ^a^	142.7 ± 18.1 ^a^
**7**	cyanidin-derivative	24.5 ± 4.0 ^b^	46.1 ± 8.7 ^a^
**8**	*	101.1 ± 23.9 ^b^	146.2 ± 25.9 ^a^
**9A** and **9B**	*	391.9 ± 7.8 ^a^	389.8 ± 76.5 ^a^
**10**	*	60.7 ± 11.1 ^a^	71.2 ± 10.4 ^a^
**Total phenols**		2583.4 ± 0.02 ^b^	3259.2 ± 0.05 ^a^

* Not identified. Different lowercase letters indicate a difference. Data are expressed as mean ± SE, *n* = 4.

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
