# Peer review of "Antioxidant Responses of Phenolic Compounds and Immobilization of Copper in Imperata cylindrica, a Plant with Potential Use for Bioremediation of Cu Contaminated Environments"

_plants, 2020, doi:10.3390/plants9101397_

Round 1

Reviewer 1 Report

The errors found in the manuscript are marked in magenta color.

Keywords

It is suggested to change the word bioremediation, since it is generally recommended that keywords are not included in the title of the manuscript.

Introduction

It is suggested to use the term potentially toxic elements instead of heavy metals, considering what is reported by Pourret and Hursthouse, 2019. doi:10.3390/ijerph16224446

Materials and methods

Page 11 lines 304-305

Indicate which is the concentration of copper sulfate that was used, it is ambiguous to say an adequate amount.

Author Response

Reviewer #1: Comments and Suggestions for Authors

Keywords

It is suggested to change the word bioremediation, since it is generally recommended that keywords are not included in the title of the manuscript.

R: Thank you. Done. The keyword “Bioremediation” was replaced by “Bioaccumulation” (line 37).

Introduction

It is suggested to use the term potentially toxic elements instead of heavy metals, considering what is reported by Pourret and Hursthouse, 2019. doi:10.3390/ijerph16224446

R: The term heavy metals (HM) was replaced by potentially toxic elements (PTE), as proposed by the reviewer throughout the document. Please, see the new documents with track changes.

Materials and methods

Page 11 lines 304-305. Indicate which is the concentration of copper sulfate that was used, it is ambiguous to say an adequate amount.

R: Thank you for your comment. We have detailed the procedure and the concentration of the copper sulfate solution used to add the Cu to the substrate. Please, see the changes in the section 4.1, lines 314 to 316. After sterilization, the sand/vermiculite mixture was supplemented with an equivalent to 200 mL of a solution 5,9 g L-1 of CuSO4·5H2O, which was allowed to equilibrate for 2 weeks at room temperature, as described by Aponte et al. [48].

Reviewer 2 Report

This manuscript deals with a comprehensive characterization of Cu responses and tolerance for a potential Cu accumulator plant. This work assessed biomass, Cu accumulation, and further identified phenolic compound that induced in response to Cu stress. Nevertheless, I have concerns about the rationale and several minor technical issues.

Regarding the term ‘antioxidant responses of phenolic compounds’ in the title, the roles of phenolic compounds act as antioxidants upon Cu or heavy metal stresses were not clearly introduced for readers. A comprehensive literature review is required to strengthen the importance and novelty of this topic. Additionally, the author emphasized that roots of Imperata cylindrica accumulated higher levels of Cu than shoots, but the assessment and identification of phenolic compounds were performed in shoots only. It would be helpful to explain their designs and reasons, so the antioxidant activity from phenolic compounds could be more likely a mechanism to cope with heavy metal stresses.

Minor issues:

  1. Which range of Cu concentration is suitable for using Imperata cylindrica as an accumulator?
  2. Line 98-99:…’ranging from 43 to 108% (Fig. 1B).’ It is not easy for me to see this range in Fig. 1B.
  3. Table 1. HCADs. This is the first time using this abbreviation. Please describe in table footnote.
  4. Line 256 ‘from’ should not be Italic.

Author Response

Reviewer #2: Comments and Suggestions for Authors

This manuscript deals with a comprehensive characterization of Cu responses and tolerance for a potential Cu accumulator plant. This work assessed biomass, Cu accumulation, and further identified phenolic compound that induced in response to Cu stress. Nevertheless, I have concerns about the rationale and several minor technical issues.

Regarding the term ‘antioxidant responses of phenolic compounds’ in the title, the roles of phenolic compounds act as antioxidants upon Cu or heavy metal stresses were not clearly introduced for readers. A comprehensive literature review is required to strengthen the importance and novelty of this topic

R: Thank you for your suggestion, we agree with you. To complement this lack of information a paragraph was added in the introduction section aimed to describe how phenolic compounds cope with the stress produced by potentially toxic elements. Please, see lines 61 to 68. “In the latter, phenolic compounds are characterized by having various functions in plants, highlighting its antioxidant capacity observed under different stress conditions [11]. Exposure to PTE generate reactive oxygen species (ROS) and increase the production of phenolic compounds in plants [12]. Phenols act chelating metals through hydroxyl and carboxyl groups and inhibit lipid peroxidation by trapping alkoxyl radical [13]. Therefore, phenolic compounds exert their effects by scavenging ROS or avoiding their generation [14,15]. Scavenging ROS activity of phenolic compounds are determined by their chemical structure, type and number of functional groups [16].”

Additionally, the author emphasized that roots of Imperata cylindrica accumulated higher levels of Cu than shoots, but the assessment and identification of phenolic compounds were performed in shoots only. It would be helpful to explain their designs and reasons, so the antioxidant activity from phenolic compounds could be more likely a mechanism to cope with heavy metal stresses.

R: We agree with your observation, but the explanation about the use of shoots for the procedure is due to the quantity of biomass and the final objective of the study. Detailing, this report is part of a PhD research project that aims to elucidates the genetic response of I. cylindrica to cope with the Cu stress. For this reason, the roots, where most of Cu is accumulated, were mainly used for the creation of the cDNA libraries, and then sequenced for transcriptomics. As a considerable amount of RNA is needed for this procedure, unfortunately only a minor fraction of root tissues was remaining being unable to perform other biochemical determinations. Despite the previous, we used the shoots for the characterization of phenolics aiming to elucidate some responses based in other metabolic pathways in this plant, which can be useful as a previous step for a most complete metabolomic characterization.

Minor issues:

Which range of Cu concentration is suitable for using Imperata cylindrica as an accumulator?

R: This question is clarified in the discussion, where a paragraph was added. Thank you for this observation. Was interesting to observe that the range in which this plant species can grow is even of most amplitude. Please, see lines 244-250. “Also, previous studies have shown how I. cylindrica inhabits soils with more than 300 mg kg-1 of Cu available [8] and how its accumulating capacity in the root increases according to the amount of Cu applied to the soil, at least between a range of 450-780 mg kg-1 of Cu in soil [35]. Although these reference values are a first approximation, the range of Cu levels in which I. cylindrica can be used in bioremediation seems to be much broader than the explored so far. Besides, data from Meier et al. [35], showed that its combined use with arbuscular mycorrhizal fungi can increase their bioaccumulation in root.”

Line 98-99: “ranging from 43 to 108% (Fig. 1B).” It is not easy for me to see this range in Fig. 1B.

R: We clarified this point. This comparison was performed based in the difference between the treatment and the control at 2 and 21 days (the extreme values). Please, see lines 100 to 102. “The exposure to Cu has a higher effect in the shoot lipid peroxidation than in roots (Fig. 1B), producing a strong increase in the MDA content for the three Cu exposure times, compared to the controls, ranging from 43% at 2 days and reaching a 108% of increase at 21 days (Fig. 1B).”

Table 1. HCADs. This is the first time using this abbreviation. Please describe in table footnote.

R: The abbreviation of HCADs was described in the footnote. Thank you for this suggestion. “HCADs: Hydroxycinnamic acid derivatives. (see Table 1) (line 160)”

Line 256 ‘from’ should not be Italic

R: Done, it was a mistake.  "from" was modified to “from” (line 266)